# Children's right to play in Chilean hospitals: A forgotten right?—A qualitative study protocol

**Aleksandra Głos**[1]*, **Alejandra Santana López**[2], **Ivonne Vargas Celis**[3],
**Lilian Sanhueza Díaz**[4], **Constanza Quezada**[2,3]

1 Center for Bioethics, Faculty of Medicine/Faculty of Law, Pontifical Catholic University of Chile, Santiago de Chile, Chile, 2 School of Nursing, Faculty of Medicine, Pontifical Catholic University of Chile, Santiago de Chile, Chile, 3 Faculty of Social, Legal and Humanities Sciences, Gabriela Mistral University, Santiago de Chile, Chile, 4 Faculty of Social, Legal and Humanities Sciences, Catholic University of Temuco, Temuco, Chile

* aleksandra.glos@uc.cl

## Abstract

### Background

Despite being a recognised human right (art. 31 of the United Nations Convention on the Rights of the Child), children's right to play is still insufficiently researched, understood and, above all, implemented. In Chile, the National Child Health Programme recognises the importance of this right for the hospitalised children but makes its provision dependent on the hospital's capacity. We therefore hypothesise that the provision of children's right to play in Chilean hospital settings will be irregular, varying from institution to institution, possibly reflecting the existing socio-economic inequalities in the country, thereby leaving much room for improvement.

### Aims

This study aims to collect qualitative data on the institutional arrangements and opportunities for children's play in public hospitals in Chile. Specifically, our goal is to (a) interpret the conditions and opportunities for play that Chilean public hospitals provide to children and adolescents; (b) interpret institutional arrangements and professional experiences of children's play in those institutions; (c) identify factors that favour and/or limit opportunities for children's play in the Chilean hospital setting.

### Methods

This is exploratory qualitative research, combining multiple and instrumental case study with ethnographic research. Its data collection techniques are semi-structured interviews with professionals in hospital settings (supported by a self-assessment questionnaire), and non-participant observations of hospital playrooms (and/or other play spaces).

**Data availability statement:** No datasets were generated or analysed during the current study. All relevant data from this study will be made available upon study completion.

**Funding:** ANID, Fondo Nacional de Desarrollo Científico y Tecnológico: Fondecyt de Iniciación en Investigación 2023: From patients to citizens. A study on narrative solidarity in bio-ethics. Grant Number: 11230025 The funders had no role in study design, data collection and analysis, decision to publish, or preparation of the manuscript.

**Competing interests:** The authors have declared that no competing interests exist.

## Conclusion

This will be the first study to investigate children's right to play in Chilean hospitals, with a particular focus on the extent of its implementation, its understanding among health professionals, as well as existing limitations and opportunities for its development.

## Introduction

### Background

**Article 31 UNCRC: A Forgotten Right in Paediatric Hospital Setting.** Children's right to play is considered a basic human capacity [1] or a basic human good [2], without which human life loses its dignity and joy. It is also a positive right, recognised by the international community and enshrined in the United Nations Convention on the Rights of the Child [3] in article 31, ratified by almost all countries (except the USA), including Chile (1990).

The UN Committee on the Rights of the Child in its General Comment to art 31 underlines the importance of children's right to play for the child's enjoyment of the highest attainable standard of health recognized in article 24 [4]. It emphasises that hospitalised children should enjoy their right to play, as it "can make an important contribution to facilitating their recovery" [4]. The importance of children's right to play for their health and well-being has also been increasingly recognised in jurisprudence and bioethical literature [5–14].

However, despite this growing legal and theoretical discussion, children's right to play is still far from being universally protected and fulfilled, especially in the sensitive hospital setting. On the contrary, it is considered the most undervalued, neglected [9] or even forgotten right [15,16]. As Hubbuck and Cross [17] argue, the hospital setting is still a play-deprived environment, despite its "special need for play" related to play's therapeutic benefits for sick and vulnerable children. A good example of this deprivation in a cross-country perspective is the recent comparative study [18] describing the State Parties' compliance with art. 31, based on the obligatory reports on their compliance with the UNCRC provided by the signatories to the Convention under article 44 and the Concluding Observations of the Committee on the Rights of the Child. This study proves that children's right to play can still be described as the "least known, least recognized and least understood" (p. 42) right of childhood. According to this study, the majority (69,23%) of countries that delivered reports do not even once mention children's right to play. From those few countries that did mention it, most report only scattered initiatives introduced to support play in the educational and social spheres, with almost no reference to the sensitive healthcare setting. Indeed, the studies on the realisation of children's right to play in hospitals are very scarce (with a few exceptions of some governmental or NGO-led reports, mainly from the UK and US [19,20], and, as far Latin American is concerned, only some preliminary considerations [21,22]. The most recent report to Committee on the Rights of the Child (CRC) presented by Chile [23] mentions the importance of play

as one of the best values of childhood (p. 21) but does not report any specific initiatives undertaken to implement the right to play, neither in general, educational nor hospital contexts. The literature review did not show any traces in the scientific literature of a similar study on children's right to play ever made in Chilean hospitals, neither in an academic nor governmental/organisational context, nor is there any systematic information on this topic.

It is worth mentioning that many paediatric hospitals and health care facilities in developed countries employ certified professionals to provide psychosocial support and interventions for children and families. For example, in the United States, as early as 1960, the American Academy of Paediatrics (AAP) [24] recommended that all paediatric wards should have a playroom equipped with appropriate materials such as games, toys and books, while the Canadian Paediatric Society in 1978 [25] advocated the employment of child life specialists to meet the psychosocial needs of hospitalised children. The acknowledgement of play was a significant step in recognizing the psychosocial needs of hospitalized children and improving patient care [26]. It resulted in the foundation of the child life profession, which is now present in the majority of paediatric hospitals in the US and Canada [27]. The child life specialists are certified professionals who use play and other psychosocial interventions to promote children's well-being and minimize the adverse effects of hospitalization for young patients. There is consistent evidence that child life services improve quality and outcomes in paediatric care as well as the patient and family experience [28,29]. This comparative perspective very clearly demonstrates the need to explore how developing countries such as Chile fulfil their obligations under UN CRC to provide play opportunities to hospitalized children, and what kind of obstacles they encounter along the way. As the study aims to identify factors that promote and limit the provision of this right, as well as collect health professionals' experience in that respect, the data gathered might help to advocate for the greater implementation of children's right to play in the context of limited resources.

**Play: Theoretical Underpinnings.** In its conceptualization of play, this project follows what has so far been its most operational definition: the one adopted in the UNCRC, but interpreted against a broader theoretical background. The Committee on the Rights of the Child, in its General Comment to Art. 31 [4], defines children's play as "any behaviour, activity or process initiated, controlled and structured by children themselves", which is "driven by intrinsic motivation and undertaken for its own sake, rather than as a means to an end" (p. 3). Play, as the Committee further states, "involves the exercise of autonomy" and its "key characteristics are fun, uncertainty, challenge, flexibility and non-productivity" (p. 3). This definition of play falls within what is described in the literature as intrinsic or cultural theories of play [9,12,30,31]. An important aspect of this framework is that it is consistent with children's understanding of play. As empirical research on this topic points out, the key feature of children's play is that it is chosen and directed by the children themselves [32]. In the educational contest, many studies report that activities organized and supervised by a teacher, even if the children admitted that they were fun, were not considered play [33,34].

In affirming the intrinsic value of play, the Committee on the Rights of the Child does not neglect its developmental (instrumental) functions. Thus, the Committee emphasises two instrumental uses of play [4, p. 4]: firstly, its importance for children's education and physical, social, cognitive and emotional development and, secondly, its relevance for children's health and well-being (broadly defined according to the WHO definition of health [35]).

According to this classification, two general types of play in a healthcare context can be distinguished 1) free play or play of intrinsic value and 2) play of instrumental value, i.e., play that is used, following the categorisation by Gjærde et al.[36], as a) a form of therapy, e.g., play activities used as rehabilitation for patients with cerebral palsy or an exercise for patients with obesity, cystic fibrosis, or asthma [37–38]; or b) play used as a preparation for medical procedures, e.g., dentist treatments [39], diagnostic tests such as injections [40, 41], blood-drawing [42] or operations [43]; c) play used in patient education, for example: a puppet used to educate patients about the treatment of their disease, such as diabetes [44] or oral hygiene [45]; and d) play as an adaptation intervention [46, 47].

However, children's free play has an irreplaceable role in ensuring children's well-being in hospital. According to Lester and Russell [48], "through playing, children situate themselves in a better state of mind–body–environment interaction". Play,

as a child's natural way of being, normalises difficult experiences [49, 50], allowing them to better adapt to the experience of hospitalisation and to transform medical institutions into places of hope and trust [51, 52]. As one child in the Capurso et al. study [53 p. 15] expressed it, being able to play with other children in the hospital made the hospital stay "not as ugly an experience as I was expecting". Furthermore, the imaginative nature of play, it's unreal "as if" [48] makes the world of play a place where children can express and process their problems and worries, e.g., the stress of being ill and hospitalised, and thus possibly regain a sense of control over this difficult situation [54]. This attests to the natural, therapeutic potential of play [49,55–57] and the importance of protecting and respecting it as a right in the sensitive healthcare setting.

Therefore, while the instrumental value of play can be very useful and beneficial in a hospital, it should not overshadow its intrinsic value. The paradox of play is that its benefit to development and well-being depends on it retaining the intrinsic value for children, that is, allowing them to maintain, as Bogatić [58] points out, the "sense of ownership of play" [59–62]. A study carried out in five different countries reinforces that, in the experience of children (both boys and girls), the most significant thing about play is the sense of agency experienced when performing the activity as an end in itself [63].

Furthermore, legally speaking, children's specific human right to play should not be violated through the instrumentalisation of play. By emphasising children's entitlement to play, the authors of this study want to underline that it should not be treated as a luxury [15], an optional after-school activity that children have to earn, but treated rather, as van Gils [64] aptly put it, as a "child's right to be a child" that should be honoured out of respect for children's dignity. Thus, the present project aims to explore children's overall opportunities for play, with a particular focus, however, on free play in the playroom (see observation guide and semi-structured interview guide, questions under item III, IV), without neglecting the instrumental importance of play, namely, the extent to which it is a tool in the hands of adults (see semi-structured interview guide, in particular questions under item II, V).

### Hypothesis and aims

Children's right to play is a recognised human right (art. 31 UNCRC). The World Health Organization Regional Office for Europe included it into the list of fundamental standards (standard 3) for fulfilment of children's rights in hospitals [65]. Moreover, for example, article 7 of the European Association for Children in Hospital [66] states that "children shall have *full opportunity* for play, recreation and education suited to their age and condition and shall be in an environment designed, furnished, staffed and equipped to meet their needs". In comparison, the Chilean National Programme for Children's Health with a Comprehensive Approach recognises children's right to play, but states that the provision of this right will *depend on "the possibilities of the hospital"* [67]. We therefore hypothesise that the fulfilment of children's right to play in Chilean hospitals will be irregular, varying according to the institution, and will depend both on its resources, as well as on management and health professionals' awareness of the importance of play – thus leaving much room for improvement. As Chile has one of the highest levels of income and opportunity inequality in the OECD countries [68], we suspect that the provision of play opportunities in Chilean hospitals (e.g., infrastructure, playroom equipment, play organisation and financing scheme) may reflect the existing socio-economic inequalities in the country.

**General aim.** The general objective of the study is to describe the institutional arrangements and opportunities for play provided to children and adolescents hospitalised in public healthcare institutions (paediatric hospitals and general hospitals with paediatric wards) in Chile.

**Specific aims.**

1. To interpret the conditions and opportunities for play that children and adolescents have in Chilean public healthcare institutions;

2. To interpret the institutional arrangements and professional experiences of healthcare professionals working with children and adolescents in Chilean healthcare institutions with regard to play;

3. To identify factors that favour and/or limit opportunities for children's play in the Chilean hospital setting.

**Setting**

The study will be conducted in public paediatric hospitals and general public hospitals with paediatric wards located in four different regions of Chile: The Metropolitan Region, the Valparaíso Region, the Arica and Parinacota Region and the Araucanía Region.

Chile is long and narrow strip of land in the southern cone of Latin America, reaching a length of 4,270 kilometres and a maximum width of 90 kilometres. It has a very heterogeneous physical geography, climate and urban development, an important geographic dispersion, and still has a large rural population, especially in the south of the country. It also has a very diverse cultural geography. The majority of Chile's population originates from a process of colonisation and a steady increase in migratory flows from Latin American countries, mainly across the country's northern border. According to the 2017 census. 12.8% of Chile's population consider themselves members of the indigenous population [69]. This gives the country a socio-cultural heterogeneity which is interesting to consider. Administratively, Chile consists of 16 different regions that vary with regards to their geographical, as well as socio-economical and intercultural characteristics. The following four regions have been selected to account for these differences in the most representative way:

1) Central Chile, the Metropolitan Region: this is the largest region in the country in terms of population density. It has a population of 7,112,808 inhabitants, of which 3,462,267 are men and 3,650,541 are women [70]. Out of the total population, 31.80% is considered to belong to indigenous groups, predominantly the Mapuche (614,881), Diaguita (9,381) and Quechua (8,366) [71]. In addition, this region is home to 61.3% of the migrant population residing in Chile [71]. On the other hand, according to the Socioeconomic Characterisation Survey [72], 4.4% of the region's population lives in poverty and 1.3% in extreme poverty.

2) Central-Coastal Zone, region of Valparaiso: As for the Valparaíso region, according to the 2017 census [70], this region has a total of 1,815,902 inhabitants, of which 880,215 are men and 935. 687 are women. Out of its total population, 5.48% is considered to belong to indigenous groups, predominantly the Mapuche (92,589), the Rapa Nui (4,566) and the Aymara (2,810), and 6.5% of the migrant population resides in the city of Valparaíso [71]. On the other hand, according to the Socioeconomic Characterisation Survey [72], 6.6% of the total population of the region lives in poverty and 1.9% in extreme poverty.

3) North of Chile, the region of Arica and Parinacota: according to the last census conducted in 2017, the total population of this region reached 226,068 people, of which 112,581 were men and 113,487 women. Out of its total population, 3.61% is considered to belong to indigenous groups, predominantly the Aymara (59,432 persons) and the Mapuche (7,858). According to the National Institute of Statistics [71], 2.2% of the migrant population in Chile resides in Arica and Parinacota. On the other hand, according to the Socioeconomic Characterisation Survey [66], 9.2% of the population of the region lives in poverty and 2.7% in extreme poverty.

4) Southern Chile, the region of Araucanía: the 2017 Census [70] shows that the population in this region amounts to 957,224 inhabitants, of which 465,131 are men and 492,093 are women. In relation to the total population [71], 14.70% is considered to belong to indigenous groups, predominantly the Mapuche (314,174), Diaguita (218) and Aymara (216). The National Institute of Statistics INE [71] reports that 1.4% of the country's migrant population resides in the region of Araucanía. According to the Socioeconomic Characterisation Survey [72], 11.6% of the population lives in poverty and 3.3% in extreme poverty, making the region the third poorest in the country.

**Study design**

This investigation falls under the framework of exploratory qualitative research. This paradigm is justified for the following reasons:

- The first is the scarcity of studies on children's play opportunities in paediatric institutions around the world, and in particular in Latin America. In this regard, the purpose of this study is to conduct preliminary explorations of the state of play in Chilean public hospitals, paving the way for future, more in-depth research.

- The second reason is related to the design of national legislation on children's play in hospitals [67], which, while respecting this right, makes it dependent on the capacity of individual institutions. This implies that the ultimate quantity and quality of play opportunities available to children in Chilean hospitals will be determined as much by the resources of the institution, as by the managemental and professional decisions made within it. This calls for entering the institution with qualitative techniques that will allow us to examine not only the resource-related, infrastructural dimension of the matter, as well as its interplay with health professionals' subjective awareness of the bioethical importance of play and the structural decisions that come with it. Qualitative research provides insight into social phenomena from within human institutions, interactions, and communications by collecting individual and group experiences [73]. To begin this line of research, it is necessary to investigate the level of awareness and decision-making of health professionals from various backgrounds (both medical and non-medical: social or psychological), who, due to their proximity to the patient, have a significant impact on children's opportunities to play in particular health institutions in Chile.

- Third, we intend to complement this knowledge through the non-participant observation of playrooms (and/or other play spaces) in public hospitals. This comparison will allow us to juxtapose the official discourses of health professionals (analysed against the backdrop of national, regional, or, if applicable, internal play policies) with the real play opportunities offered to children in these institutions.

More specifically speaking, to realise the objectives of the study, we will use two data collection instruments. First is the interview, which allows for the collection of data or information from the subject of study through verbal interaction with the researcher [74]. This instrument will be used to learn about the health professionals' levels of awareness of the importance of play in hospital settings, as well as their decisions and experiences in this field. We have opted for semi-structured interviews, as this technique "has a greater degree of flexibility than structured interviews, because they are based on planned questions that can be adjusted to the interviewees" [75, p.163] (see S1 File).

We will complement the standard qualitative procedure of a semi-structured interview with a non-participant observation of the playroom (and/or other play spaces). In this context, ethnographic design serves as a useful technique for "understanding the relationship between actors' perspectives on meaning and actions in the situational circumstances in which they occur" [76, p. 99]. In this way, we aim to determine how a specific place, in this case a playroom (and/or other play spaces), operates and whether it is appropriate for its users. Non-participant observation can provide useful information not only on the space's infrastructure, accessibility, equipment, and its functionality (see S2 File).

This exploratory-descriptive framework will be based on the constructivist paradigm [77, 78], which emphasises the importance of open-ended approaches, recognises the cultural context of data collection, and acknowledges the irreducible subjectivity of both participants and researchers. It will be complemented by the multiple case study design [79], which is structured around instrumental and collective cases (Fig 1).

1) We will work under an *instrumental* modality, since we seek to understand a general issue: the institutional arrangements and opportunities for play available to hospitalised children in Chile. In particular, by exploring the subjective experience in paediatric health institutions gathered through the interviews with different actors, we aim to reveal the implicit meanings that determine the play opportunities in particular institutions and investigate their correlation with the real-life play opportunities provided to children in those facilities, as observed in playrooms (and/or other play spaces) in the ethnographic portion of the study.

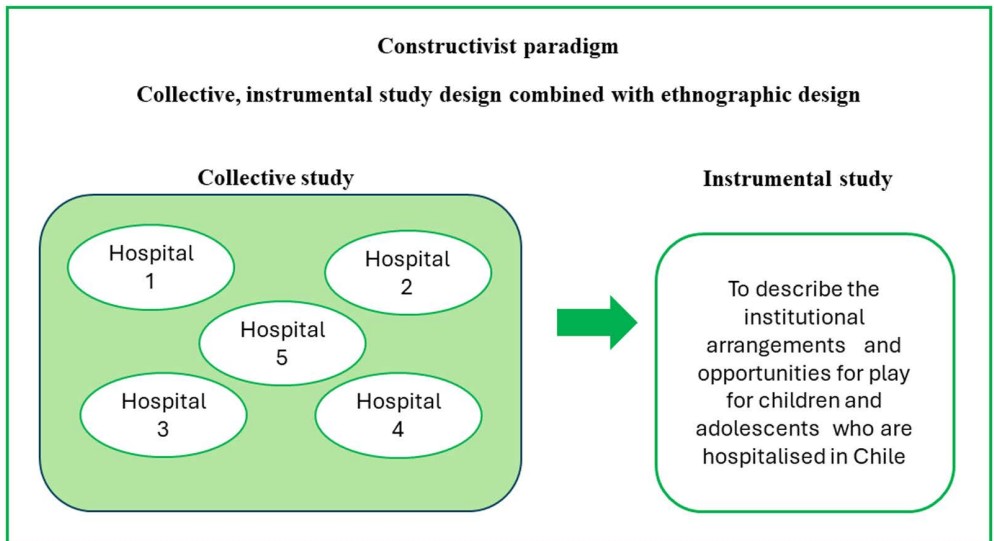

**Fig 1. Study design.**

2) In addition to its instrumental modality, this study fits into a *collective* case study framework. We will collaborate with five Chilean paediatric hospitals in various locations to provide a representative case for their regional categories. This will allow us to account for variability by representing regional features, socioeconomic status, and intercultural differences.

## Materials and methods

### Sample, sample selection criteria, data saturation criteria

As indicated in the previous section, the cases will be gateways to particular socially constructed worlds in the paediatric hospitals/wards. Cases were selected according to attributes of similarity and difference, but also on the basis of the feasibility of accessing them, taking as a reference the theoretical saturation at the level of the cases and at the level of the participants, without detriment to the above. A *minimum* parameter of participating cases (5) and participating persons (15) was established to safeguard the validity of the study.

With regards to the key informants, the convenience sampling strategy will be used, i.e., "a population is chosen, but it is not known how many subjects may have the phenomenon of interest, resorting to the subjects that are found" [80, p. 1149]. In a complementary manner, the principles of theoretical saturation sampling will be taken into account, indicating an initial purposive sampling of informants who possess the phenomenon, which is analysed in successive phases, incorporating new informants according to the emerging categories with a purpose of giving solidity to the categories and theories of the field of study [80].

**Inclusion criteria for hospitals.** In particular, the selection of hospitals (cases) was determined on the basis of the following inclusion criteria:

- Chilean public hospitals

- Paediatric hospitals or general hospitals with paediatric wards

- Hospitals located in the Metropolitan Region (geographic/administrative centralisation)

- Hospitals located in regions (geographic/administrative decentralisation)

- Hospitals located in urban areas

   **Inclusion criteria for participants.** The selection of informants is determined by their key role in the administration or use of children's playrooms (and/or other play spaces) in the hospital setting:
The following inclusion criteria are defined:

- Health professionals working in paediatric hospitals or on paediatric wards of general hospitals

- Health professionals who make decisions regarding the allocation and management of the use of space for children to play within the hospital

- Other hospital professionals who design technical or administrative guidelines related to the rights of children and adolescents as patients and can address the issue of the right to play.

- Other professionals working with children and promoting their well-being in paediatric hospitals or on paediatric wards of general hospitals

- Professionals who promote the development of play as part of initiatives that favour the quality of life of hospitalised children.

The research team had no prior contact with the selected institutions, which were chosen solely on the basis of their geographical location (being representative of each region) and characteristics (being a paediatric hospital or a general hospital with paediatric wards).

## Data collection

The multiple case study approach guided the selection of data collection methodologies. The data collection instruments to be used are: an individual semi-structured interview guide (see S1 File) aimed at managers, administrative staff, physicians, psychologists, occupational therapists, social workers, nurses, and/or others (complemented with a self-assessment guide based on the WHO [65] guidelines: "Children's Rights in Hospital: Rapid Assessment Checklist": S3 File); a guide to non-participant observation of playrooms (and/or play spaces) in hospital settings (see S2 File).

1) Semi-structured individual interviews will be conducted using a questionnaire, which will allow for a broad and flexible collection of these actors' professional and situated perspectives on institutional arrangements and play opportunities for hospitalised children and adolescents. The interview guide has been developed based on a literature review and study objectives, and it includes both key questions and potential follow-up questions. This design will allow the interviewer to delve deeper into the interviewee's narrative and/or establish new themes that arise during the conversation. (See S1 File).

2) Non-participant observation guideline. Standardised observation times will be carried out in the hospital's playroom (and/or other play spaces) for children's play. It will follow the guideline (see S2 File) developed by the research team for this study, based on its objectives, ethnographic methodology selected, and considering the experience of other research in the field of play [81–84]. Non-participant observation, also known as non-intrusive observation, is a research technique in which the observer does not interact directly with the subjects or participants of the study. In this case, the observer will not interfere with the environment or the activities being observed.

3) Self-assessment guide is the additional tool to complete the interviews with key informants. It is based on the WHO [66] guidelines: "Children's Rights in Hospital: Rapid Assessment Checklists" translated by the research team into Spanish. It will be a supplementary instrument used to systematise the information obtained during the interview and estimate the grade of implementation of children's right to play in hospitals according to the globally followed guidelines regulating this matter (see S3 File).

The data collection process will be subject to the conditions set by the healthcare institutions. The study team will contact hospital staff and coordinate instances of observations and interviews according to their availability and hospital capacity, while attempting not to disrupt their job activities.

The below table summarises the relationship between the specific objectives, the categories of analysis and the dimensions to be explored.

Data collection is planned to be carried out in two stages, as described below (Fig 2):

Given the flexibility and recursiveness of qualitative designs [85,86], phases one and two do not have to be linear and can potentially be implemented simultaneously if health institutions allow it.

In the *first phase of data collection*, team members will conduct semi-structured interviews with key informants who have agreed to take part in the study. Interviews will be conducted by the members of the research team (composed of scholars of the following disciplines: two social workers, an ethicist, an anthropologist and a lawyer with an interdisciplinary second degree), four of whom have expertise and several years of experience in conducting qualitative interviews. The best-qualified member of the research team will train (according to the guidelines developed by Flick [87] the fifth research team member, whose practical field experience is limited. Interviews will be conducted in person or via zoom depending on the preference of the interviewee and their health institutions. The Arica and Parinacota regions of northern Chile were important to include in the study, but interviews will only be conducted via zoom, as no member of the research team is a resident of this region. These will be recorded and then transcribed to ensure the accuracy of what was reported by the participants. This phase will be supplemented by an additional tool in the form of a self-assessment guide, which will be emailed to interview participants who have already agreed to it during the interview.

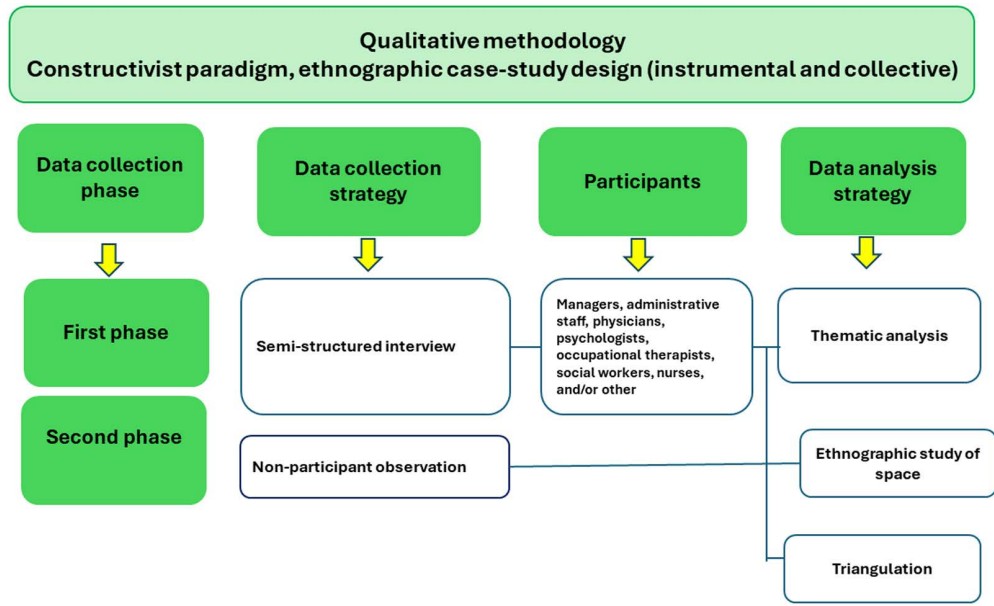

**Fig 2. Extended study design.**

In the *second phase of data collection*, non-participant observations of the play rooms and/or other play spaces in the hospitals will be carried out. The observation will be carried out by a group of four research assistants (advanced anthropology students), each with previous experience of field observation in research projects. This work will be supervised by an academic anthropologist from the research team, who will provide field training in the observation process, taking handwritten notes in a field notebook and photographs, and subsequent transcription of findings [88,89,90]. The anthropologist will supervise the process and evaluate each field note transcription based on the observation guidelines developed by the research team, indicating in each case their completion if necessary, to ensure the quality and consistency of the results. Observations will take place at the time indicated by each institution (to ensure that children are not present). Observers will be introduced by the person designated to look after the project at each hospital and presented to relevant individuals working in the institutions. In order to be more rigorous with the information collected, each space will be observed at least three times at 1-hour intervals on different days, and will be supplemented as necessary. The period of one hour of observation of the space allows us to see changes in it, for example if there are changes in light or if there some other than play activities conducted in the playroom [84]. In relation to the changes that the space may undergo, it was decided to go on different occasions (min. two different days). Every member of the project involved will be remunerated and all travel expenses will be covered.

## Data analysis

To analyze the data, we will employ Stake's [79] case study interpretation technique and deal with cases in a transversal manner. This means that we will evaluate individual cases in conjunction with the examination of the set of cases in general. For hospitals analysed as cases, interpretations will initially focus directly on the data collected (the corpus of interviews and observation records) at individual institutions. Subsequently, the analysis will be applied to the set of cases, considering their similarities, differences and particularities of five hospitals, allowing the meanings of the set of cases to be captured.

More specifically, the analysis process will be organised as follows:

**1. The semi-structured interviews analysis.**

In this research, the categorisation process will be developed following the guideline of thematic analysis model [91, 92]. Preliminary categories of analysis are constructed deductively, prior to data collection, as reflected in S1 File. They are pre-established on the basis of contributions from scientific literature on the importance of children's right to play in health or hospital contexts, and the also reflect the CRC [4] distinction to intrinsic value of play for children's well-being, as well its instrumental value and possible uses by the health professionals. This deductive will be completed by the inductive openness to the emergence of other categories arising from this analysis. After the coding process, the categories will be adjusted according to the contributions of the data collected. This analytical approach is typical for qualitative designs that do not seek generalisations of knowledge, but rather the transferability of knowledge generated in previous studies, with an intentional openness to the particular knowledge of the socio-cultural contexts in which the research is carried out (openness to emerging categories) [93].

The analysis will follow a six-phase process, as outlined by Braun and Clarke [91]. Specifically, the data will be transcribed by a professional who will sign the confidentiality agreement, and will then be anonymized by the principal investigator. All transcripts will be checked back against the audio recordings by the principal investigator. All researchers will code openly according to pre-established codes based on the research objectives, as well as new codes of an emergent nature that respond to the particularities of the contexts in which the data will be collected. The research team will hold an initial meeting to review the pre-established codes, agree on shared definitions and their precision parameters and potential attributes, in order to facilitate further analysis. Codes will be then collapsed or expanded, and the initial codebook will be produced by each author individually. The initial themes and subthemes developed by each author will then be

triangulated, refined, discussed and conceptualized by the whole research team through team meetings, in which potential biases will be controlled and the interpretation of the data will be enhanced in a collective and interdisciplinary manner. The analysis and triangulation of data will be done by the research team manually. Additionally, the Atlas Ti software will be used for (1) document identification (transcription), (2) transfer of codifications carried out manually and (3) generation of code reports and related citations. The usage of the software will be instrumental, secondary and limited in application to complement and systematize in conjunction with the interdisciplinary analysis conducted manually by the research team. To enhance trustworthiness, we will apply Braun and Clarke's 15-point checklist of criteria for thematic analysis to our study [91].

## 2. The non-participant observation analysis.

The study objectives will guide the collection and analysis of this data, with non-participant observation informing on the opportunities that the playroom and/or other play spaces provide for children to exercise their right to play in hospitals. In particular, we want to examine to what extent the observed space with its material culture favours or limits play opportunities in these institutions. For this aim, a descriptive analysis of place will be conducted. This analysis will focus solely on the material aspect of play space (its infrastructure, equipment, aesthetics, design, location, usability). As such, it will look at space and objects "from a perspective of data assemblage, where other data sources are investigated that can give insights into a broader picture of life and human activity" [94, p.621]. The findings of this analysis will allow us to verify, supplement and correct the information obtained through the process of interviews with key informants [95].

In accordance with Martínez [76], an approach of exclusivity and inclusivity will be adopted: a) exclusivity, because the key elements to be observed are predetermined, as described in the observational guideline (Appendix 2), and b) inclusivity, which will allow for the inclusion of elements that emerge from the context, so that the observation is more comprehensive and realistic, capturing more fully what is happening in the examined environment. In this way, the units of meaning that may emerge in the analysis of the information recorded will faithfully correspond to the reality observed in each hospital.

The analysis will be carried out on the basis of field notes taken by the researchers during non-participant observations (and subsequently transcribed). The field notes will be organised by means of predetermined tables on the observation guideline. In non-participant observation, the observer's task is to contemplate what is happening in a distanced way [76], describing as accurately as possible all relevant objects and their materiality, the arrangement of furniture and artefacts, the environment etc. From these records, each observer will draw up a classification system for each observed space, to be discussed and subsequently interpreted by the research team.

Once sufficient observation data is collected, we will begin an analysis of each playroom and/or other play space in the hospital to identify play opportunities for children in this institution, and then compare the analysis of each case with the others. In the process of analysis, we will aim to establish the relationships between the place and its objects with data on play programs and dynamics, collected through semi-structured interviews. The findings will be organised through tables, diagrams, interspersed with written text, which will be included in the published results [90]. The detailed timeline of the study is described in the table below (Fig 3).

## 3. Triangulation of results.

Triangulation is a procedure used to ensure the reliability of qualitative research aimed at contrasting different perceptions of the researched phenomenon, leading thus to consistent and valid interpretations [73,96,97]. It consists of the combination of different methods, study groups, local and temporal settings and theoretical perspectives when dealing with a phenomenon [73]. We have chosen to triangulate methods and data collection techniques [98] for this study. First, we will work with a constructivist study design which combines case study and ethnographic methods. Secondly, we have opted for two types of data collection techniques, the semi-structured interview and non-participant observation. This framework

is complemented by an interdisciplinary triangulation of researchers [98], related to the fact that the research team includes investigators from within different disciplines, which will allow us to detect or minimise possible biases of individual researchers and enrich our understanding of the topic through a comprehensive, interdisciplinary analysis. As the research team comprises a legal scholar with interdisciplinary background, an anthropologist, social workers and an ethicist, the study will combine, respectively, legal expertise in children's rights and in particular children's right to play, interpreted against the wider interdisciplinary background with anthropological perspectives on play and its material culture, and psychosocial conditions associated with the lives of children and their environments in clinical setting. Researchers from different disciplines will be equally involved in data analysis and triangulation (contributing from their perspective to a broad and comprehensive understanding of the research phenomenon), while being able to enrich the in-depth analysis of its different parts (for example, the presence of an anthropologist will ensure the quality of the ethnographic analysis of the place, a lawyer will ensure the correct analysis of the data in light of legal and public policy aspects, while social workers will be able to provide insights into the professional dynamics in health care and their impact on play, as well as enhance the understanding of children's living conditions related to the hospital context).

Regarding the process of methodological triangulation, the research team will first examine the convergence or divergence of what was established through the interviews with key actors and the observation of the play spaces in individual hospitals. Secondly, we will consider how these convergences or divergences provide information on the actual opportunities and/or barriers to children's play in each of the studied hospitals. Finally, individually analysed cases (each serving as a representative case for its regional category) will be compared with others in order to draw conclusions that will contribute to a better understanding of the general phenomenon, that is, the reality of the Chilean public hospitals in terms of their implementation of children's right to play.

## Safety and ethical considerations

This study complies with the CIOMS International Ethical Guidelines for Health-related Research Involving Humans [99] and the ASA Ethical Guidelines updated in 2021[100]. The study has been approved by the UC Health Sciences Scientific Ethics Committee and the written resolution certificate was obtained (18th April 2024, Ref 231218019) and shall observe national regulations (Law 20.120) [101] for any changes occurring throughout the process.

Although this study deals with children's right to play in hospital settings, only adults (hospital professionals) will be interviewed and non-participant observations will be made of the play spaces in the selected public hospitals. Our researchers will not engage with vulnerable population. Considering the non-participant nature of ethnographic observation, undue interference will be avoided when conducting observations of playrooms and/or play spaces. [100]

The publicly available contact details of the hospitals will be used in order to access the institution, present the research and invite the chosen institutions to participate. If the institutions accept the invitation, their involvement will be formalised with a letter of authorization, which is required by Chilean national law governing human-research-related activities (art. 10 of Law 20.120 [101]). Once the institutional authorizations are in place, contacts will be made to identify possible participants, who will then go through a process of invitation and information about the research, which, if successful, will result in the signature of the informed consent document.

The study will only involve those individuals who, after explaining the aim and method of the study, voluntarily agree to participate, for which their written consent will be obtained. The informed consent document will provide contact information for the principal investigator and the Research Ethics Committee that has approved this study, so that participants can ask questions at any time. In the same document, participants will be informed that they are free to terminate their participation in the study at any time.

Participants have the right to privacy, confidentiality and anonymity. Therefore, transcribed interview and non-participant observation data will be coded and used for research purposes only. Records verifying the participant's identity (name, national number, physical and email address and telephone number) will remain confidential even after the publication of

this study. This study will ensure that no information to be published will allow the identification of the persons or places that participated in the research.

Research data will be kept in a locked cabinet and in a cloud folder accessible only to authorised researchers. All documents containing research data will be stored on a computer storage device that can only be accessed with a separate password.

In the event that an individual decides to drop out of the study or withdraw their consent to participate, they will be immediately excluded from the study and no further data will be collected. Data collected in these instances will not be used for the study and will be destroyed.

## Data Management

As the study will gather information through the use of two strategies of data collection: (1) semi-structured interviews with key informants, complemented by a self-assessment questionnaire, and 2) non-participatory observation of playrooms and/or other spaces in selected hospitals, the data management strategy will be presented for each of these separately:

1) The information obtained through the semi-structured interviews will be collected via Zoom recordings that the participants will have agreed to through informed consent. A specialised team will transcribe these verbatim, and the research team will then anonymise them in a first review. Once an anonymised transcript of each interview is available, the Zoom file will be removed. Interview transcripts will be arranged by application date (E[Nº]_DDMMYYY) and named accordingly. In addition, a database is considered, with each file clearly identifying the participant, the duration of the interview, the interview style (face-to-face/online), interviewer, and any pertinent comments. Informed consent will be filed in a folder that will be kept in the custody of the PR. The self-applied questionnaire data will be saved in a shared folder by the core team and sorted by key informant interview identification (C[Nº]_DDMMYY).

2) The information collected through observations will be arranged using the filled observation guidelines developed by the research team. The guidelines files will be organised by their application site, observation number, and collection date (H[Nº]_OBS01_DDMMYYY). Furthermore, the creation of a database is considered, with each file carefully labelled with the observation site, duration of the observation, research assistant, and any relevant comments.

These documents will be stored in a locked cabinet and in a cloud folder with limited access and shared exclusively with the core research team. This folder will be administered by the PR. In addition, as a precaution, the PR will keep and back up an external hard disc containing the data. All data collected over the course of the research will be kept for five years after it concludes.

The collected data will be made available through its dissemination in a qualitative data repository (e.g., National Research and Development Agency Repository/ Repositorio Agencia Nacional de Investigación y Desarrollo [ANID] repository https://www.re3data.org/repository/r3d100014225 or other). As regards non-participant observations, transcription of anonymised observation field notes will be published. In the case of semi-structured interviews, the code book and extracts from the transcripts will be published to exemplify the application of the codes. For this, the informed consent document explicitly states that the data obtained could be shared in repositories of this type.

## Other

### Dissemination of results

The results of the study will be disseminated through standard academic channels, including publications in peer-reviewed scientific journals, as well as presentations at national and international conferences. In addition, we plan to disseminate the project results to the communities participating in the study. Typically, qualitative projects involve the creation of a feedback space between researchers and communities, where preliminary findings are presented and the community

has the opportunity to participate in the analysis, correct an area that identifies them or has been misinterpreted by the researchers, and highlight areas of interpretation that have been more neglected and that they value. This will take the form of a one-day seminar with the community participating in the study, during which we intend to create a space for dialogue and reciprocal learning. The project will conclude with a public conference in which the research team will present the study's findings and invited speakers will deliver talks on related subjects.

### Study limitations

Considering that this is an exploratory study, and, to our knowledge, the first of its kind in the region, it limits itself to learning about objective conditions and subjective factors that determine the provision of right to play to hospitalised children in Chile. As such, it does not investigate children's experiences, opinions, and eventual claims with respect to their own right. Such an investigation is, in our opinion, crucial, but would require a separate study, conducted with different methodological tools, an expanded research team (qualified to interview children) and extreme caution due to the engagement with the most vulnerable subjects. This is why this critical aspect must be the focus of future research. For the same reason, and due to the institutional limitations, we could not include observation of children at this research stage, limiting ourselves to the observation of play spaces and their equipment. However, both aspects will be addressed in future research.

As it is a study of limited resources, it is impossible to investigate all 16 regions of the country. However, the regions selected for the study represent in our opinion the geographical and socio-cultural heterogeneity of the country very well, and as such, may lead to very interesting findings and possibilities for future research.

Finally, the fact that in this study the variables of cultural and gender diversity are not directly considered can be interpreted as a limitation. Some authors argue that the cultural variable should be included in children's play theory [102], which is especially important in multicultural countries like Chile, which has a significant presence of rural and indigenous populations as well as a steady increase in migratory influxes. For example, Alonqueo Boudon et al. [103] report that, in rural Mapuche contexts, play does not tend to be competitive, with group interaction amongst children of different ages predominating, even when boys and girls are observed to ignore each other. Given the diversity of the regions in which the hospitals participating in the study are located, the preliminary findings may provide insights into possible cultural differences in understanding the importance of play and its provision, paving the way for future research that examines cultural and gender variables more directly.

### Supporting information

**S1 File. Semi-structured interview guide.**
(DOCX)

**S2 File. Non-participant observation guide.**
(PDF)

**S3 File. Self-assessment guide.**
(PDF)

### Author contributions

**Conceptualization:** Aleksandra Glos, Ivonne Vargas Celis.

**Funding acquisition:** Aleksandra Glos.

**Methodology:** Aleksandra Glos, Alejandra Santana López, Ivonne Vargas Celis, Lilian Sanhueza Díaz, Constanza Quezada.

**Writing – original draft:** Aleksandra Glos, Alejandra Santana López, Ivonne Vargas Celis, Lilian Sanhueza Díaz, Constanza Quezada.

**Writing – review & editing:** Aleksandra Glos.

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
