## [Decision Letter · Decision Letter 0]

22 Jan 2025

PONE-D-24-19286Children’s Right to Play in Chilean Hospitals. A Forgotten Right? – A Qualitative Study ProtocolPLOS ONE

Dear Dr. Glos,

Thank you for submitting your manuscript to PLOS ONE. After careful consideration, we feel that it has merit but does not fully meet PLOS ONE’s publication criteria as it currently stands. Therefore, we invite you to submit a revised version of the manuscript that addresses the points raised during the review process.

Comments from the editorial office: Upon internal evaluation of the reviews provided, we kindly request you to disregard the reviewer report provided by Reviewer 1. No amendments are required in response to Reviewer 1’s comments

We look forward to receiving your revised manuscript.

Kind regards,

Annesha Sil, Ph.D.

Staff Editor

PLOS ONE

“ANID, Fondo Nacional de Desarrollo Científico y Tecnológico: Fondecyt de Iniciación en Investigación 2023: From patients to citizens. A study on narrative solidarity in bioethics. Grant Number: 11230025

Reviewers' comments:

Reviewer's Responses to Questions

**Comments to the Author**

1. Does the manuscript provide a valid rationale for the proposed study, with clearly identified and justified research questions?

Reviewer #1: Yes

Reviewer #2: Yes

Reviewer #3: Yes

2. Is the protocol technically sound and planned in a manner that will lead to a meaningful outcome and allow testing the stated hypotheses?

Reviewer #1: Yes

Reviewer #2: Yes

Reviewer #3: Yes

3. Is the methodology feasible and described in sufficient detail to allow the work to be replicable?

Reviewer #1: Yes

Reviewer #2: Yes

Reviewer #3: No

4. Have the authors described where all data underlying the findings will be made available when the study is complete?

Reviewer #1: Yes

Reviewer #2: Yes

Reviewer #3: Yes

5. Is the manuscript presented in an intelligible fashion and written in standard English?

Reviewer #1: Yes

Reviewer #2: Yes

Reviewer #3: Yes

6. Review Comments to the Author

You may also provide optional suggestions and comments to authors that they might find helpful in planning their study.

Reviewer #1: Children’s right to play in Chilean Hospitals. A Forgotten Right? A Qualitative Study Protocol: Reviewers Comments

The rationale of the Study

The proposed study on children’s right to play is relevant within the context of Chilean public hospitals. The inquiry focuses on the institutional arrangements and opportunities for play available to children and adolescents who are hospitalised in public hospitals, an area that has received limited research attention, thus addressing an important gap in research.

This research will contribute to understanding the significance of mapping and enhancing play facilities in pediatric wards and advocating for children's right to play. The findings will benefit various disciplines beyond medicine, such as occupational therapy, social work, psychology, and physiotherapy. It is intended that the findings of the proposed study be used to enhance the play opportunities for children and adolescents in Chilean public hospitals. This proposed study has the potential to be replicated in hospitals globally, including private hospitals.

The rationale of the study is clear and valid. The aim and the objectives are feasible.

Theoretical Framework

The Play theoretical framework is suitable for the study. However, it is not explicitly stated which play theory is being used. It would therefore be beneficial for the authors specify and clearly articulate the theory they are employing, such as Huizinga's Cultural Play Theory, Stuart Brown's Theory, or others. Additionally, considering the integration of Locke's theory of human rights with the play theory could further enhance the theoretical framework as it relates to the rationale of the proposed study.

Methodology

Taking a qualitative approach using the constructive paradigm is well justified. Using a design that is exploratory design is appropriate. The location of the study is also suitable and well-justified.

To enhance the study, it is recommended that the authors consider using a mixed-method approach to incorporate and account for the rapid assessment checklist they plan to use, which is quantitative. This will provide a more comprehensive and balanced analysis of the data.

Sampling

The convenience and purposive sampling techniques are adequately described and justified. The selection process and criteria for the participants and hospitals are clear.

Issues to be addressed by authors are:

• To state the exclusion criteria for both participants and hospitals.

• Clarify which region will have two cases and the justification for that decision.

• The authors should include a section on how they will gain access to hospitals and how the ethical approval processes are carried out in Chilean public hospitals. Discuss the approval process in depth, including whether the Ministry of Health or hospitals themselves approve research.

Data Collection and tools

Semi-structured interviews, a non-participant observation guide, and a rapid assessment checklist are appropriate tools for collecting data in this study. The data collection process is described in detail.

Issues to be addressed by authors are:

• Will the semi-structured interview schedule and the NPO guide be piloted? If not, provide a justification.

• The authors should explain why they have chosen online data collection over face-to-face interviews.

• The use of Ethnographic methodology is mentioned, but clarity and justification are needed regarding the duration of observations. The NPO tool can be used as a stand-alone tool.

• Will the interviews be conducted in English or Spanish?

• Rapid Assessment Checklist:

It is unclear who in the pediatric ward or hospital will be recruited to administer the assessment. Please clarify.

The “WHO” assessment checklist covers seven standards for assessment. The translated checklist covers Standard No.1 from numbers 1 to 21. Is this intentional? If so, please justify.

Data Analysis

The data analysis processes are clearly outlined. If the mixed method approach recommendation is adopted, the analysis will have to be adjusted accordingly.

Trustworthiness and Rigour

The authors have explained how they will use triangulation to ensure the trustworthiness and rigor of the data.

• It is important to include all aspects of trustworthiness, such as credibility, transferability, dependability, and confirmability.

• Consider the validity and reliability of the Rapid assessment.

Ethical Considerations

The ethical considerations are well-described, but a discussion on beneficence is necessary.

Timeline of the study

The timeline needs to be revised for accuracy.

Dissemination of results

The authors have outlined a comprehensive plan for disseminating the study results, which includes:

• Publishing in peer-reviewed scientific journals

• Presenting at conferences

• Conducting a seminar for study participants

• Organizing a public conference by the research team

Reviewer #2: The proposed study is very good and the study design is apprpriate and well described. I wish the investigators all the very best for this study.

Reviewer #3: Dear authors, I appreciate the opportunity to review your proposal.

I would like to highlight that the proposal has significant potential to contribute to the advancement of knowledge and its implementation in developing countries such as Chile. The proposal is well-structured, clear, and addresses a relevant topic with a robust scientific approach. Additionally, it objectively presents the theoretical context of the research, emphasizing the importance of the right to play for hospitalized children and justifying the relevance of the study in the Chilean context. However, a few aspects could be improved or clarified to further strengthen the quality of the proposed research.

1. In the introduction, it is pertinent to highlight the reality of developed countries, such as the United States and Canada, which differ significantly from South American countries due to the presence of professionals known as Child Life Specialists. These professionals focus on using play to promote the well-being of hospitalized children. Comparing these distinct realities could further emphasize the relevance of the proposed study, demonstrating that, regardless of the existence of a professional dedicated to this role, play should be recognized and promoted as a fundamental right of every child.

2. Although the project defines free play, chosen and directed by children, as a guiding principle of the process, it is important to emphasize that playful interactions between children and healthcare professionals are also valuable sources of data. These interactions can either promote or restrict play, depending on the actions of the professionals. Including an analysis in the interviews regarding how professionals approach play and their attitudes toward playful activities is insightful, but it could be further enriched if this analysis is also incorporated into the observation phase. For example, observing situations where professionals encourage or inhibit play (due to lack of time, perceptions of the value of play, or institutional factors) could reveal important dynamics. It is suggested to integrate these interactions into the observation process, considering the comparison between the narratives in the interviews and the actions observed in the environment, which are not always aligned.

3. The study's hypothesis is solid and addresses a real and potentially significant issue. Its approach offers the potential to reveal valuable insights into the disparities in upholding the right to play for hospitalized children in Chile.

4. The research setting is well-represented, and the selection of the four regions is appropriate to capture the geographical, socioeconomic, and cultural diversity of Chile, which may impact the guarantee of hospitalized children’s right to play.

5. The choice of a qualitative approach is appropriate as it allows for a deeper understanding of the factors influencing children’s opportunities to play in hospitals. This approach enables the exploration not only of the material and structural resources of healthcare institutions but also of the perceptions and decisions of healthcare professionals regarding play, which is essential for understanding practical barriers and opportunities. The focus on the different perspectives of healthcare professionals, including both medical and non-medical staff, is a relevant approach, as play can be influenced by various factors within the hospital. The comparison between interview narratives and observational d

6. Regarding the materials and methods, further clarification is needed on how the interviews will be conducted and the composition, roles, and qualifications of the research team. Will the research team be present in the various regions selected in the country? Are they residents of these regions? Will there be financial resources allocated for the team's travel? Who will conduct the interviews? Does the interviewer have expertise in qualitative interviews? Will there be training provided? Do they have any prior connection to the research setting? Additionally, it remains unclear whether the interviews will be conducted in person or remotely.ata will provide a robust foundation for understanding the phenomenon under study.

7. The systematic approach to observation needs to be better described. Who will be the observer? Do they have experience with this method of data collection? Will training be provided on how to document the findings from the observations and ensure consistency among observers? How much time, in hours or days, is planned for the observations? Will they occur during different shifts? Will the individuals working in the observed settings be informed about this proposal? If behaviors are being observed, I recommend consulting the Ethics Committee regarding the need to obtain informed consent (IC), even if prior approval has been granted. When observing behaviors, particularly in a context involving interactions with children, it is essential to ensure that informed consent is obtained. While the proposed non-participant observation might involve minimal or no interaction with individuals, it still requires informed consent, especially given the vulnerability of the child population.

8. I understand that the lack of information regarding how the research team will be composed, trained, and how the logistics for conducting observations in different hospitals will be handled is a significant gap in the proposal. In terms of resources, it would be crucial for the project to detail in the methodology whether there will be adequate financial support or logistical backing to ensure that the team has the necessary resources to travel, conduct observations, and collect data systematically. This would ensure the feasibility of the research and its potential for replication.

9. The proposal for analyzing the interviews and observation data is well-structured and follows a solid qualitative analysis approach. The use of a combination of deductive and inductive analysis is relevant, especially since the study is based on an in-depth and contextualized understanding of the phenomena, rather than seeking generalizations. Furthermore, the use of Atlas Ti software as a tool to facilitate the organization and coding of interviews is an effective choice. However, it would be interesting to provide more detail on how the integration between manual analysis and the software will be carried out. The idea of an interdisciplinary analysis also seems important, but could be more specified in terms of how different areas of expertise within the team will contribute to the analysis. Nevertheless, the proposal for triangulation is well-applied, relevant, and, in this context, provides clarity and depth to the research findings.

10. In summary, the study is promising and addresses a highly relevant topic with the potential to positively impact hospital practices related to the right to play. The suggestions presented aim to further strengthen the proposal, ensuring greater clarity and feasibility for its implementation.

7. PLOS authors have the option to publish the peer review history of their article (what does this mean? ). If published, this will include your full peer review and any attached files.

**Do you want your identity to be public for this peer review?** For information about this choice, including consent withdrawal, please see our Privacy Policy .

Reviewer #1: **Yes: ** Babalwa Dano

Reviewer #2: **Yes: ** Manoj Shankarrao Patil

Reviewer #3: No

---

## [Author Response · Author response to Decision Letter 1]

3 Feb 2025

Dear Reviewer 3,

Thank you very much for reviewing our work and for providing such insightful, detailed comments. We found the advice friendly and the criticism constructive. According to your suggestions, we made the following changes:

I would like to highlight that the proposal has significant potential to contribute to the advancement of knowledge and its implementation in developing countries such as Chile. The proposal is well-structured, clear, and addresses a relevant topic with a robust scientific approach. Additionally, it objectively presents the theoretical context of the research, emphasizing the importance of the right to play for hospitalized children and justifying the relevance of the study in the Chilean context. However, a few aspects could be improved or clarified to further strengthen the quality of the proposed research.

● Comment: 1. In the introduction, it is pertinent to highlight the reality of developed countries, such as the United States and Canada, which differ significantly from South American countries due to the presence of professionals known as Child Life Specialists. These professionals focus on using play to promote the well-being of hospitalized children. Comparing these distinct realities could further emphasize the relevance of the proposed study, demonstrating that, regardless of the existence of a professional dedicated to this role, play should be recognized and promoted as a fundamental right of every child.

Response:

Thank you for this comment. Indeed, the idea of the study emerged from the comparison between the Chilean reality and the far-reaching implementation of children's right to play in hospitals in developed countries, where there is not only the practice but also a longtime, ongoing academic discussion on this topic, which is still very rare in the Latin American context. We are very grateful for this comment, as it is indeed important to express this comparative perspective more clearly. Moreover, this comparison will help us to build argumentation to promote the greater recognition of children's right to play in hospitals, which we intend to do through this research (and its follow-up). The long experience of the United States and Canada and the evidence-based effectiveness of child life specialist practice can, mutatis mutandis, serve as a model for other countries.

Changes in the manuscript: We added the following fragment (please check p. 3-4)

It is worth mentioning that many paediatric hospitals and health care facilities in developed countries employ certified professionals to provide psychosocial support and interventions for children and families. For example, in the United States, as early as 1960, the American Academy of Paediatrics (AAP)24 recommended that all paediatric wards should have a playroom equipped with appropriate materials such as games, toys and books, while the Canadian Paediatric Society in 197825 advocated the employment of child life specialists to meet the psychosocial needs of hospitalised children. The acknowledgement of play was a significant step in recognizing the psychosocial needs of hospitalized children and improving patient care26. It resulted in the foundation of the child life profession, which is now present in the majority of paediatric hospitals in the US and Canada27. The child life specialists are certified professionals who use play and other psychosocial interventions to promote children’s well-being and minimize the adverse effects of hospitalization for young patients. There is consistent evidence that child life services improve quality and outcomes in paediatric care as well as the patient and family experience 28,29. This comparative perspective very clearly demonstrates the need to explore how developing countries such as Chile fulfil their obligations under UN CRC to provide play opportunities to hospitalized children, and what kind of obstacles they encounter along the way. As the study aims to identify factors that promote and limit the provision of this right, as well as collect health professionals’ experience in that respect, the data gathered might help to advocate for the greater implementation of children’s right to play in the context of limited resources.

● Comment 2. Although the project defines free play, chosen and directed by children, as a guiding principle of the process, it is important to emphasize that playful interactions between children and healthcare professionals are also valuable sources of data. These interactions can either promote or restrict play, depending on the actions of the professionals. Including an analysis in the interviews regarding how professionals approach play and their attitudes toward playful activities is insightful, but it could be further enriched if this analysis is also incorporated into the observation phase. For example, observing situations where professionals encourage or inhibit play (due to lack of time, perceptions of the value of play, or institutional factors) could reveal important dynamics. It is suggested to integrate these interactions into the observation process, considering the comparison between the narratives in the interviews and the actions observed in the environment, which are not always aligned.

Response: That’s a very astute remark. Our focus on free play is due to two main reasons; first is the fact that the intrinsic, non-instrumental (albeit with countless benefits) value of play is less recognized in the, by definition, utility-oriented hospital setting in general; secondly, it captures the fact that play is also the right of children. However, it is very true that medical play is not less important, as it has multiple benefits of reducing children's anxiety before the procedures, establishing a trusting relationship with healthcare providers and preventing medical trauma, among others. It is also true that professionals’ attitude has an enormous impact on play provision, and it may indeed “encourage or inhibit” play. Health professionals might not only engage (or not) in the medical play with children but also promote or inhibit their free play. We fully agree that it would be very valuable to observe these interactions and attitudes in clinical settings. Unfortunately, we are not able to include this aspect in the observation guidelines at this stage because we cannot yet (in this study) observe children, having to limit ourselves to observation of the spaces (playrooms, if they exist or other play spaces, their design, equipment and other aspects). This is because hospitals in Chile are very close, restricted institutions, adopting strict measures of child protection and security. Additionally, to the evaluation by the Ethics Committee of our university, the project had to go through the Ethics Committee of each institution (two of which were already obtained, additionally to two exemptions). As this is the first, exploratory, diagnostic study on the topic, we have decided to establish our relations with the hospitals in a step-by-step manner. In this study, we want to let ourselves be known to the hospitals through our good work and ethical conduct, so that in the second step (and the second study) we are also permitted to interview children and observe their interactions with medical professionals during play. Children are the main protagonist of our study, and it is an undeniable limitation that we cannot include them at this stage. However, the study is part of a larger project and a long-term commitment of the research team, so in the next study we will incorporate this helpful suggestion of the Reviewer to a greater extent. What we have been able to do at this stage to honor this suggestion is to make the following changes:

1) We included more questions about the medical play and professional interaction dynamics of children in the semi-structured questionnaire. As the study design includes interviews with a diverse group of professionals (both of medical and non-medical specialization), we can get a broad picture of medical play, as well as its beneficial and inhibiting factors.

2) We also explicitly addressed this particular limitation in the Study Limitations section.

Changes in the manuscript and supplementary files:

Ad. 1. Interview guide

We added additional questions to our semi-structured interview guide (point 5).

V. Medical play

1. How do healthcare professionals use play when working with children? Do they seem to understand the importance of play in their practice?

a) Specification, if necessary: for example, a nurse or other professional using play to prepare children for painful or frightening procedures, such as MRIs or injections or doctors using play when performing a procedure or explaining the medical condition.

b) Have they received training, are they supported by additional staff?

2. In your professional practice, were there any cases (some particular situations with paediatric patients) that made you understand the importance of this type of play, for example, as an information strategy, rehabilitation, therapy, etc.)? Could you tell us more about it?

3. It is sometimes said that ‘play is a language’ of children. Do you consider that play can be a more effective form of communication with children? How can play facilitate communication with children? (Ask for some examples)

4. Play is not equally valued by all adults or health professionals. In your practice, have you observed the interactions of medical professionals with children that promote or restrict play? In what sense? How does that happen? Why?

Ad. 2 Study Limitations (p. 17)

We added the following passage (please check p. 17):

For the same reason, and due to the institutional limitations, we could not include observation of children at this research stage, limiting ourselves to the observation of play spaces and their equipment. However, both aspects will be addressed in future research.

● Comment: 3. The study's hypothesis is solid and addresses a real and potentially significant issue. Its approach offers the potential to reveal valuable insights into the disparities in upholding the right to play for hospitalized children in Chile.

Response: Thank you!

● Comment 4. The research setting is well-represented, and the selection of the four regions is appropriate to capture the geographical, socioeconomic, and cultural diversity of Chile, which may impact the guarantee of hospitalized children’s right to play.

Response: Thank you!

● Comment 5. The choice of a qualitative approach is appropriate as it allows for a deeper understanding of the factors influencing children’s opportunities to play in hospitals. This approach enables the exploration not only of the material and structural resources of healthcare institutions but also of the perceptions and decisions of healthcare professionals regarding play, which is essential for understanding practical barriers and opportunities. The focus on the different perspectives of healthcare professionals, including both medical and non-medical staff, is a relevant approach, as play can be influenced by various factors within the hospital. The comparison between interview narratives and observational

Response: Thank you!

● Comment: 6. Regarding the materials and methods, further clarification is needed on how the interviews will be conducted and the composition, roles, and qualifications of the research team. Will the research team be present in the various regions selected in the country? Are they residents of these regions? Will there be financial resources allocated for the team's travel? Who will conduct the interviews? Does the interviewer have expertise in qualitative interviews? Will there be training provided? Do they have any prior connection to the research setting? Additionally, it remains unclear whether the interviews will be conducted in person or remotely.

Response: Thank you for this comment. Yes, we did not explicitly touch upon these aspects so that could remain unclear. The research team resides in 2 of the chosen regions. As the longitudinal distances in Chile are very short, in contrast to latitudinal distances, conducting interviews will be possible in person in the 3rd, and via zoom in the 4th (Northern Chile). Observation in each case will be conducted in person, as one of the students engaged in the project is originally a resident of the Arica y Parinacota region studying in Central Chile, the Metropolitan Region (the student will be remunerated and her travel cost will be covered). In any case, with the exception of the Arica y Parinacota region, we will combine in person and remote interviewing mode according to the preference and agenda of particular institutions and interviewees. The interviews will be conducted by members of the research team: four of whom (two social workers, one anthropologist, and one ethicist with qualitative experience) have several years of experience in doing qualitative interviews. The fifth member (a lawyer with interdisciplinary second degree) has a solid theoretical foundation and limited field experience (gained in the parallel ongoing project), and will therefore receive supplemental training by the best-qualified member of the research team. The whole team is totally impartial and is composed of academics having no prior connection to the research setting. We have completed the manuscript as regards the missing information, as can be seen below:

Changes in the manuscript:

a) Sample, sample selection criteria, data saturation criteria (page 10)

The research team had no prior contact with the selected institutions, which were chosen solely on the basis of their geographical location (being representative of each region) and characteristics (being a paediatric hospital or a general hospital with paediatric wards).

b) Data collection (page 11)

In the first phase of data collection, team members will conduct semi-structured interviews with key informants who have agreed to take part in the study. Interviews will be conducted by the members of the research team (composed of scholars of the following disciplines: two social workers, an ethicist, an anthropologist and a lawyer with an interdisciplinary second degree), four of whom have expertise and several years of experience in conducting qualitative interviews. The best-qualified member of the research team will train (according to the guidelines developed by Flick87 the fifth research team member, whose practical field experience is limited. Interviews will be conducted in person or via zoom depending on the preference of the interviewee and their health institutions. The Arica and Parinacota regions of northern Chile were important to include in the study, but interviews will only be conducted via zoom, as no member of the research team is a resident of this region. These will be recorded and then transcribed to ensure the accuracy of what was reported by the participants. This phase will be supplemented by an additional tool in the form of a self-assessment guide, which will be emailed to interview participants who have already agreed to it during the interview.

Comment: 7. The systematic approach to observation needs to be better described. Who will be the observer? Do they have experience with this method of data collection? Will training be provided on how to document the findings from the observations and ensure consistency among observers? How much time, in hours or days, is planned for the observations? Will they occur during different shifts? Will the individuals working in the observed settings be informed about this proposal?

If behaviors are being observed, I recommend consulting the Ethics Committee regarding the need to obtain informed consent (IC), even if prior approval has been granted. When observing behaviors, particularly in a context involving interactions with children, it is essential to ensure that informed consent is obtained. While the proposed non-participant observation might involve minimal or no interaction with individuals, it still requires informed consent, especially given the vulnerability of the child population.

Response: Thank you for pointing that out. We have now better

---

## [Decision Letter · Decision Letter 1]

20 Feb 2025

Children’s Right to Play in Chilean Hospitals. A Forgotten Right? – A Qualitative Study Protocol

PONE-D-24-19286R1

Dear Dr. Glos,

We’re pleased to inform you that your manuscript has been judged scientifically suitable for publication and will be formally accepted for publication once it meets all outstanding technical requirements.

Kind regards,

Roberto Ariel Abeldaño Zuñiga

Academic Editor

PLOS ONE

Additional Editor Comments (optional):

Reviewers' comments:

Reviewer's Responses to Questions

**Comments to the Author**

1. Does the manuscript provide a valid rationale for the proposed study, with clearly identified and justified research questions?

Reviewer #2: Yes

Reviewer #3: Yes

2. Is the protocol technically sound and planned in a manner that will lead to a meaningful outcome and allow testing the stated hypotheses?

Reviewer #2: Yes

Reviewer #3: Yes

3. Is the methodology feasible and described in sufficient detail to allow the work to be replicable?

Reviewer #2: Yes

Reviewer #3: Yes

4. Have the authors described where all data underlying the findings will be made available when the study is complete?

Reviewer #2: Yes

Reviewer #3: Yes

5. Is the manuscript presented in an intelligible fashion and written in standard English?

Reviewer #2: Yes

Reviewer #3: Yes

6. Review Comments to the Author

You may also provide optional suggestions and comments to authors that they might find helpful in planning their study.

Reviewer #2: The proposed study is very good and the study design is apprpriate and well described. I will be happy to see the final results. I wish the investigators all the very best for this study.

Reviewer #3: Dear Authors,

I would like to congratulate you on the careful revisions made to the manuscript, which have greatly strengthened the work. The changes implemented, particularly those related to the interview guide with the additions about free play and medical play, are extremely valuable. These inclusions have deeply enriched the data collection, providing a more comprehensive understanding of the phenomenon being studied. It is with great pleasure that I contribute to the reading of this project, which will undoubtedly have a significant impact on understanding the dynamics of play in hospital settings and how it affects the well-being of Chilean children. The way you addressed the suggestions received, clearly explaining the decisions not to incorporate some of them, was also exemplary.

Sincerely,

7. PLOS authors have the option to publish the peer review history of their article (what does this mean? ). If published, this will include your full peer review and any attached files.

**Do you want your identity to be public for this peer review?** For information about this choice, including consent withdrawal, please see our Privacy Policy .

Reviewer #2: **Yes: ** Dr. Manoj S. Patil

Reviewer #3: **Yes: ** Edmara Bazoni Soares Maia

---

## [Editor Report · Acceptance letter]

PONE-D-24-19286R1

PLOS ONE

Dear Dr. Glos,

I'm pleased to inform you that your manuscript has been deemed suitable for publication in PLOS ONE. Congratulations! Your manuscript is now being handed over to our production team.

Kind regards,

on behalf of

Dr. Roberto Ariel Abeldaño Zuñiga

Academic Editor

PLOS ONE